# Tokenize the World into Object-level Knowledge to Address Long-tail Events in Autonomous Driving

**Ran Tian**[1,2] **Boyi Li**[1,2] **Xinshuo Weng**[1] **Yuxiao Chen**[1] **Edward Schmerling**[1] **Yue Wang**[1,3]
**Boris Ivanovic**[1] **Marco Pavone**[1,4]

[1]NVIDIA  [2]UC Berkeley  [3]University of Southern California  [4]Stanford University

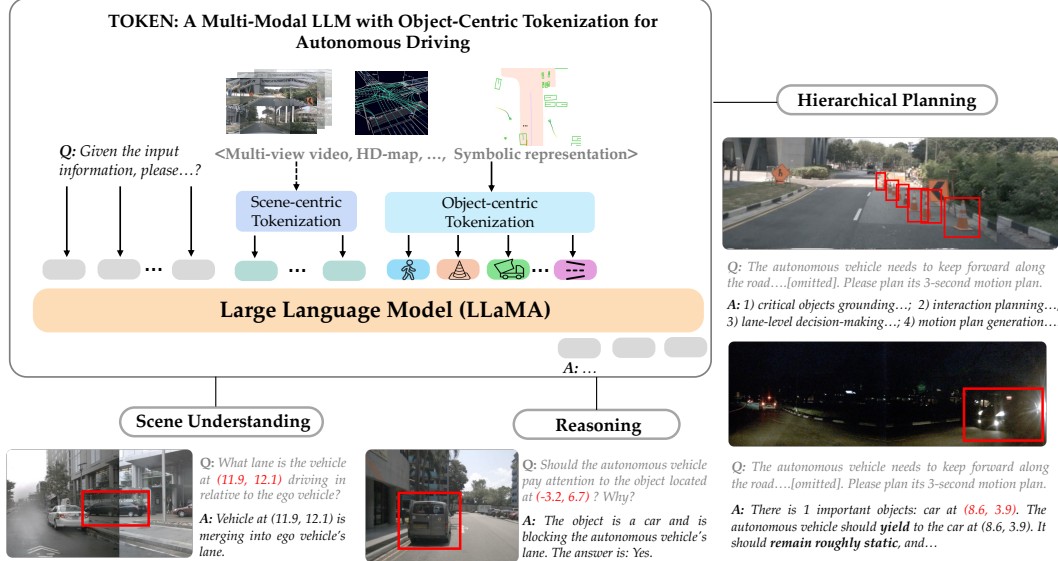

Figure 1: TOKEN is a novel Multi-Modal Large Language Model (MM-LLM) that tokenizes the world into object-level knowledge, enabling better utilization of LLM's reasoning capabilities to enhance autonomous vehicle planning in long-tail scenarios.

**Abstract:** The autonomous driving industry is increasingly adopting end-to-end learning from sensory inputs to minimize human biases in system design. Traditional end-to-end driving models, however, suffer from long-tail events due to rare or unseen inputs within their training distributions. To address this, we propose TOKEN, a novel Multi-Modal Large Language Model (MM-LLM) that tokenizes the world into object-level knowledge, enabling better utilization of LLM's reasoning capabilities to enhance autonomous vehicle planning in long-tail scenarios. TOKEN effectively alleviates data scarcity and inefficient tokenization by leveraging a traditional end-to-end driving model to produce condensed and semantically enriched representations of the scene, which are optimized for LLM planning compatibility through deliberate representation and reasoning alignment training stages. Our results demonstrate that TOKEN excels in grounding, reasoning, and planning capabilities, outperforming existing frameworks with a 27% reduction in trajectory L2 error and a 39% decrease in collision rates in long-tail scenarios. Additionally, our work highlights the importance of representation alignment and structured reasoning in sparking the common-sense reasoning capabilities of MM-LLMs for effective planning. More details at the project website.

**Keywords:** Multi-modal LLM, Autonomous Driving, Representation Alignment

8th Conference on Robot Learning (CoRL 2024), Munich, Germany.

# 1 Introduction

The autonomous driving industry is increasingly pursuing end-to-end learning from sensory inputs to reduce human inductive bias in system design [1, 2]. Despite the remarkable progress, end-to-end models inherently suffer from severe performance degradation in long-tail scenarios. For example, state-of-the-art end-to-end autonomous driving planners often fail to navigate temporary construction sites and react too aggressively to jaywalkers; even simple rule-based planners can significantly outperform high-capacity end-to-end models in these long-tail scenarios [3]. This motivates recent efforts to fine-tune Large Language Models (LLMs) into autonomous vehicle planners [4, 5, 6], aiming to leverage the benefits of both high-capacity models and the common-sense reasoning abilities that emerge from world-knowledge training.

LLM-based planners, in their simplest form, depend on textual scene descriptions as prompts, making their performance highly reliant on the quality and detail of these descriptions. Detailed prompts require extensive engineering and generate many tokens for the LLM to process. Conversely, our evaluations show that simple, heuristic prompts do not tap into the common-sense reasoning abilities of LLMs due to insufficient scene understanding. As a result, Multi-Modal Large Language Models (MM-LLMs), which naturally integrate various data modalities beyond text, are emerging as promising foundations for developing autonomy stacks in autonomous vehicles.

The predominant approach is to leverage pre-trained encoders (typically pre-trained using visual-text alignment) to extract features from the sensory inputs, followed by a querying transformer that uses latent queries to tokenize the features into dense latent tokens and feed them to the LLMs [7, 8, 9, 10, 11]. Training an effective scene tokenizer (encoder and querying transformer) often requires a significant amount of question-answer pairs (QAs) (for example, Flamingo used more than one billion QAs to reach satisfactory performance [12]). However, current MM-LLM datasets for autonomous driving typically contain fewer than one million QAs [7, 13]. Consequently, without careful model and training scheme designs, these models often exhibit poor performance in reasoning and planning tasks due to a lack of scene understanding and grounding capability. The key challenge is to enable the scene tokenizer to extract informative and structured information that can unlock the common-sense reasoning ability of the LLM in a low-data regime.

We propose **TOKEN** (Fig. 1), a novel MM-LLM framework that utilizes object-centric tokenization to tokenize the world into a few object-level tokens to enhance the planning ability of autonomous vehicles, especially in long-tail scenarios. Our key insight is that **object-level latent tokens, with each token representing a relevant object in the scene, are much more informative and easier for the LLM to interpret compared to unstructured dense tokens**. TOKEN not only produces a condensed and semantically informed representation of the scene but also enables us to use a state-of-the-art end-to-end driving model as the pre-trained scene tokenizer, effectively alleviating both the data scarcity and inefficient tokenization challenges present in current MM-LLM frameworks.

In Sec. 5.1, we compare TOKEN to alternative MM-LLM frameworks and demonstrate its superior grounding, reasoning, and planning capabilities in a low-data regime. In Sec. 5.2, we compare TOKEN to the state-of-the-art end-to-end (SOTA) autonomous driving planner [2] and showcase its strong performance in long-tail scenarios, including navigating around construction sites, executing 3-point turns, resuming motion after a full stop, and overtaking parked cars through the oncoming lane. In Sec. 5.3, we compare TOKEN to the SOTA LLM-based planner [5] and demonstrate its superiority in long-tail scenarios. We further conduct an ablation study to highlight the importance of proper representation alignment and structured reasoning process alignment in effectively evoking the common-sense reasoning ability of the LLM backbone for planning. To the best of our knowledge, we are the first to conduct an in-depth analysis to demonstrate the promising potential and necessity of such alignment in effectively leveraging MM-LLM to mitigate long-tail challenges.

# 2 Related Work

**End-to-End Driving & Long-tail Event Mitigation**. Traditional end-to-end autonomous driving models inherently suffer from performance degradation in long-tail scenarios [1, 2]. To mitigate this issue, previous works focused on detecting such situations online [14, 15, 16, 17] and switching

the planner to a model-based planner [18, 19] to ensure safety. In contrast, we leverage a pre-trained end-to-end driving model to tokenize the scene and share features with a LLM, leveraging its common-sense reasoning ability to enhance planning performance in long-tail scenarios.

**LLM-Based Motion Planning**. In light of LLMs' outstanding understanding and generalization abilities, previous works have attempted to fine-tune LLMs into motion planners [4, 5, 6, 3]. However, LLM-based planners are highly dependent on the quality and resolution of the scene descriptions. While a more comprehensive and fine-grained description can help the LLM better parse and understand the scene, it also makes the LLM inefficient to train and run inference. Additionally, designing templates to textualize scenes requires extensive prompt engineering. To alleviate the model from text-based scene description, recent works exploits object-centric vectors using parsed symbolic information [20]. However, relying on parsed symbolic information making the system less robust to inductive bias and vulnerable to salient information from the scene context [21]. In contrast, we directly uses raw sensory inputs to enhance the LLM's scene understanding ability.

**Multi-Modal Large Language Models**. MM-LLMs have been increasingly integrated into the autonomous driving stack (e.g., 3D detection [22], driving co-pilot [9], driving scene summarization [23], ). Previous works typically fine-tune existing MM-LLMs (e.g., [24, 25, 26]) using driving-related QAs. The existing MM-LLMs are mostly optimized for visual understanding. Consequently, autonomous driving MM-LLMs fine-tuned using these models [7, 8, 9, 10, 11] often lack the grounding, 3D understanding, and behavior reasoning abilities. To mitigate this issue, recent works attempt to integrate a detection head into the querying transformer (the latent queries used for token extraction additionally interact with the detection queries to guide the tokens to capture 3D information) [27] or leverage a BEV encoder pre-trained in driving tasks to extract features for the querying transformer [28]. However, the tokens in both works remain unstructured and entangled (i.e., one object's information could be distributed across multiple tokens), leading to ineffective tokenization. Our work differs from existing ones in two key ways: first, we use object-centric tokenization and a pre-trained end-to-end driving model to tokenize the scene into structured and disentangled tokens; second, we analyze the importance of representation alignment and structured reasoning process alignment to effectively evoke the common-sense reasoning ability of the MM-LLM for planning. More related work on object-centric representation in robotics are discussed in App. A

## 3 TOKEN Framework

We propose a novel MM-LLM framework, TOKEN, tailored for autonomous driving. It consists of three modules: a scene tokenizer that tokenizes the sensory inputs into object-level tokens, an adapter that aligns the object token's embedding space with the text embedding space, and an LLM.

**Object-Centric Scene Tokenization.** Existing MM-LLMs' scene tokenizers typically leverage pre-trained vision encoders (e.g., CLIP [7, 8, 9, 10, 11]) to extract features from the sensory inputs, followed by a querying transformer [25] that uses learnable queries to tokenize the features into dense latent tokens and feed them to the LLM. Training an effective scene tokenizer poses a significant challenge. On one hand, the scale of existing question-answering MM-LLM datasets for autonomous driving is far from enough. On the other hand, relying on latent queries to extract tokens in a low-data regime often results in unstructured and redundant tokens that are difficult for the LLM to interpret and process efficiently, which leads to poor performance in reasoning and planning tasks due to a lack of scene understanding and grounding capability (Sec. 5.1).

Driving is a highly structured and object-oriented task - the ego vehicle's behavior is largely constrained by the traffic agents and map elements around it. Instead of training the driving scene tokenizer from scratch through question-answering tasks and using unstructured tokens to encode the driving scene, we obtain object-centric tokens from existing end-to-end autonomous driving stacks trained on tasks such as detection, tracking and segmentation and are thus already optimized to encode rich spatial, temporal, and semantic information directly associated with relevant objects in the scene. This approach allows us to leverage state-of-the-art multi-modal end-to-end driving models as our scene tokenizer, effectively addressing both the data scarcity and inefficient tokenization challenges present in current MM-LLM frameworks. We illustrate our general framework in

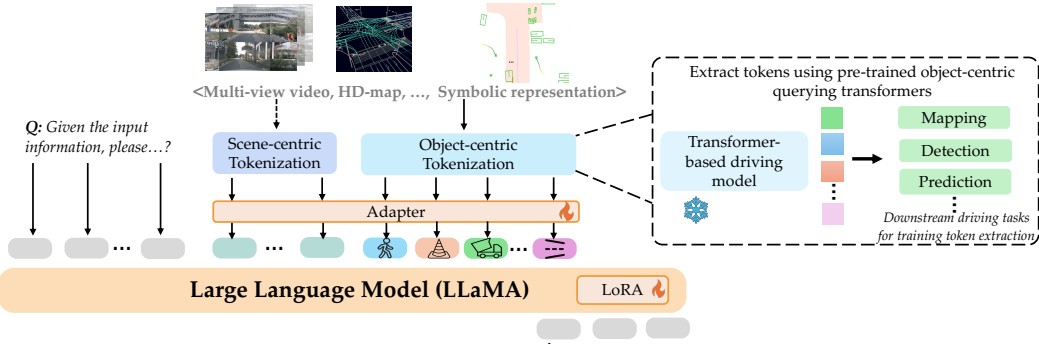

Figure 2: TOKEN obtains object-centric tokens from existing end-to-end autonomous driving stacks and uses a condensed and semantically-informed representation to encode the scene.

Fig. 2 and introduce the specific design choices of our scene tokenizer in Sec. 4.1. In addition to the object-level tokens, unstructured scene-level latent tokens learned from scratch can be optionally included to compensate for missing information, such as weather conditions.

**Adapter.** While the object-level tokens generated by our scene tokenizer already encode rich information, it is crucial to align the latent token embedding space with the text embedding space in order for the LLM to understand and extract information. We perform a wide range of QA tasks to train the adapter to align the tokens, paving the road for the subsequent behavior planning task.

## 4 Experimental Setup

### 4.1 Design Choice of the Scene-Tokenizer

In this work, we leverage the transformer-based end-to-end driving model, PARA-Drive [2], as our scene tokenizer to extract object-level tokens. PARA-Drive is a parallelized modular autonomous vehicle stack that encompasses a diverse set of modules for the co-training of bird's-eye view (BEV) features from multi-view video input. Each module is a querying transformer that uses latent queries to attend to the BEV features and decode the corresponding task output. We pre-train PARA-Drive on object-centric and scene-centric tasks, including mapping, object tracking, occupancy prediction, and motion prediction, as shown in Fig. 3. In object-centric tasks, each query produces a latent token that encodes information about a specific scene object.

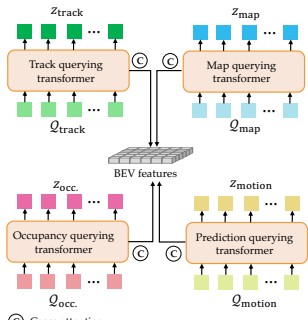

Figure 3: End-to-End driving model (PARA-Drive) as the scene-tokenizer.

Specifically, the track query $\mathcal{Q}_{\text{track}}^i$ is trained to produce a token $z_{\text{track}}^i \in \mathbb{R}^{1 \times 256}$ that encodes object $i$'s 3D bounding box and semantic category, the motion query $\mathcal{Q}_{\text{motion}}^i$ is trained to produce a token $z_{\text{motion}}^i \in \mathbb{R}^{1 \times 256}$ that encodes object $i$'s potential dynamic behavior, and the map query $\mathcal{Q}_{\text{map}}^j$ is trained to produce a token $z_{\text{map}} \in \mathbb{R}^{1 \times 256}$ that encodes the map element $j$'s geometry and semantics (e.g., crossing area) information. We concatenate the track token and motion token to constitute non-map object tokens: $z_{\text{agent}}^i = z_{\text{track}}^i \bigoplus z_{\text{motion}}^i$, and directly use the map token $z_{\text{map}}^j$ as the map element token. Although we don't use latent tokens from the occupancy prediction module, we still include this module during training to fairly compare with PARA-Drive. We follow the same training procedure in [2] to train the scene tokenizer. All non-map element tokens share one MLP adapter and all map element tokens share another one.

### 4.2 Dataset Construction

To train and evaluate TOKEN, we construct a dataset based on the NuScenes dataset [29], including visual question-answering (QA) pairs that span the full stack of autonomous driving development. We illustrate a few examples of these QAs in Fig. 1; more can be found in App. B.

**Perception.** We build perception QAs based on the DriveLM dataset [7], covering object semantics and dynamic behavior identification. Additionally, we create object-lane association QAs to enhance the model's understanding of map elements and objects' topological relationships to the ego vehicle.

**Behavior Reasoning.** Behavior reasoning QAs include two types of questions: 1) object level behavior analysis questions where we ask the model to reason about whether an object is critical (i.e., is likely to influence the ego vehicle's planning) and provide the corresponding reason; 2) scene-level critical object grounding questions where we ask the model to predict the locations of critical objects in the ego vehicle's local frame. Behavior reasoning QAs aim to enable the model to understand context-dependent critical objects, thereby connecting the object tokens with the LLM to simplify the scene for downstream planning.

**Route-conditioned Hierarchical Planning.** Different from previous works that use the relative position of the ground-truth ego trajectory to define high-level commands ("keep forward" and "turn right/right"), we re-labeled the NuScenes dataset to use road-level navigation signals as high-level commands, including: "keep forward along the current road," "prepare to turn right/left at the next intersection," "turn right/left at the intersection," "left/right U-turn," and "left/right 3-point turn." We utilize chain-of-thought reasoning to align the model's planning process and guide it to progressively generate the driving plans in three steps. First, the model identifies the critical objects in the current driving scene, including their categories and 2D locations in the ego frame. Next, it proposes the desired behavior mode, detailing interaction plans with the critical objects (e.g., overtake) and lane-level decisions (e.g., left lane change). Finally, it generates a 3-second motion plan (6 waypoints).

### 4.3 Training Strategy

We use LLaMA-2-7B [30] as our LLM backbone. We keep the scene tokenizer frozen and use a Low-Rank Adaptation (LoRA) module [31] to fine-tune the LLM. We note that the tokenizer is trained on the NuScenes dataset, ensuring that both the tokenizer pre-training and the MM-LLM training share the same source driving training data. We train TOKEN in three stages: pre-training, reasoning fine-tuning, and planning fine-tuning. More details are in App. C.

## 5 Experimental Results

### 5.1 On the Value of Object-Centric Tokenization

**Experiment Design.** We compare TOKEN against existing MM-LLMs to demonstrate its effectiveness and efficiency in scene understanding, grounding, and planning. Our experimental setup focuses on the impact of 1) different tokenization schemes and 2) pre-training on driving data.

**Baseline.** We compare TOKEN against (1) VILA-1.5 [32], which uses a trainable ViT to tokenize frames, (2) Video-LLaMA [26], which leverages a pre-trained and frozen ViT to extract visual features and a querying transformer to produce visual tokens from the extracted features, and (3) BEV-TOKEN, a variant of TOKEN that directly uses dense bird's-eye view (BEV) features from the same pre-trained PARA-Drive instead of sparse object-level tokens, similar to most MM-LLMs (e.g., Video-LLaMA). We follow the same training recipe for all models and use the same LLaMA-2-7B as the LLM backbone. For Video-LLaMA and VILA-1.5, we use the past four front view frames to render a 2-second video as input for each QA.

**Metrics.** We use classification accuracy to measure the model's ability to 1) classify the object at a location in the ego frame, 2) identify the lane in which the object is located, and 3) answer other categorical questions in the DriveLM dataset [7]. To evaluate the model's ability to localize and reason about critical objects, we use precision and recall to measure its grounding ability (we use Hungarian matching to match the predictions with the ground truth), and use accuracy to measure its ability to identify whether an object is critical given the object's center location in the ego frame. We consider three variants of trajectory L2 error to evaluate the predicted motions from different perspectives in various scenarios: the overall, turning, and progress errors, which are calculated from the original L2 distance, heading difference, and longitudinal-weighted L2 distance between the prediction and GT[1]. We use the average collision rate over the entire horizon to measure the safety of a motion plan. More details about our evaluation protocol can be found in App. D.

---

[1]We report the errors at 1s, 2s, and 3s, the averaged error over these time steps (denoted as $Ave_{123}$), and the averaged error over the entire horizon (denoted as $Ave_{all}$)

| | Scene understanding ↑ | | | Critical object grounding ↑ | | | Traj L2 (m) ↓ | | | | | Collision (%) ↓ |
|---|---|---|---|---|---|---|---|---|---|---|---|---|
| Method | Obj. class. | Lane-object asso. acc. | Acc. | Precision | Recall | Import. class. | 1s | 2s | 3s | Ave$_{123s}$ | Ave$_{all}$ | Ave$_{all}$ |
| Video-LLaMA | 0.28 | 0.39 | 0.38 | 0.22 | 0.27 | 0.58 | 0.27 | 1.72 | 6.34 | 3.01 | 2.39 | 2.64 |
| VILA-1.5 | 0.37 | 0.22 | 0.42 | 0.19 | 0.16 | 0.55 | 0.28 | 1.56 | 4.41 | 2.09 | 1.66 | 1.98 |
| BEV-TOKEN | 0.68 | 0.64 | 0.61 | 0.58 | 0.62 | 0.76 | 0.39 | 1.01 | 2.02 | 1.14 | 0.96 | 0.39 |
| TOKEN | **0.92** | **0.68** | **0.76** | **0.87** | **0.76** | **0.92** | **0.26** | **0.71** | **1.47** | **0.81** | **0.68** | **0.15** |
| PARA-Drive | NA | NA | NA | NA | NA | NA | 0.23 | 0.68 | 1.50 | 0.80 | 0.66 | 0.19 |

Table 1: **Quantitative evaluation of the scene understanding, critical object grounding, and planning tasks.** TOKEN significantly outperforms baseline VLMs due to its use of driving-task pre-trained features and object-centric tokenization. Bold numbers denote the best results in each column, and the numbers shaded in green indicate significant improvements. We also show the PARA-Drive's planning performance as reference (shaded in grey).

| | | Traj L2 (m) ↓ | | | | | Heading L2 (rad) ↓ | | | | | Lon. weighted traj L2 (m) ↓ | | | | | Collision (%) ↓ |
|---|---|---|---|---|---|---|---|---|---|---|---|---|---|---|---|---|---|
| Long-tail | Method | 1s | 2s | 3s | Ave$_{123s}$ | Ave$_{all}$ | 1s | 2s | 3s | Ave$_{123s}$ | Ave$_{all}$ | 1s | 2s | 3s | Ave$_{123s}$ | Ave$_{all}$ | Ave$_{all}$ |
| 3-point turn | PARA-Drive | 0.50 | 1.38 | 2.76 | 1.55 | 1.29 | 0.40 | 0.83 | 1.03 | 1.48 | 0.90 | 0.78 | **2.10** | **4.19** | **2.36** | **1.96** | 5.33 |
| | TOKEN | **0.39** | **1.29** | **2.60** | **1.43** | **1.18** | **0.21** | **0.35** | **0.71** | **0.42** | **0.36** | **0.68** | 2.15 | 4.33 | 2.39 | 1.98 | **4.00** |
| | TOKEN * | 0.20 | 0.73 | 1.77 | 0.90 | 0.73 | 0.35 | 0.41 | 0.37 | 0.38 | 0.33 | 0.37 | 1.28 | 2.93 | 1.53 | 1.24 | 0.00 |
| Resume motion from full stop | PARA-Drive | 0.14 | 0.79 | 2.30 | 1.08 | 0.85 | 0.11 | 0.45 | 0.57 | 0.38 | 0.35 | 0.26 | 1.46 | 2.66 | 1.38 | 1.56 | 0 |
| | TOKEN | **0.13** | **0.70** | **1.58** | **0.80** | **0.65** | **0.09** | **0.24** | **0.31** | **0.22** | **0.19** | **0.24** | **1.24** | **2.66** | **1.38** | **1.13** | 0 |
| Overtake | PARA-Drive | 0.27 | 0.89 | 1.94 | 1.03 | 0.85 | 0.05 | 0.11 | 0.18 | 0.11 | 0.10 | **0.50** | 1.56 | 3.37 | 1.81 | 1.50 | 2.3 |
| | TOKEN | 0.29 | **0.77** | **1.63** | **0.90** | **0.74** | **0.04** | **0.07** | **0.11** | **0.07** | **0.09** | 0.53 | **1.36** | **2.86** | **1.58** | **1.31** | **0** |
| Construction zone | PARA-Drive | 0.38 | 0.93 | 1.65 | 0.99 | 0.84 | 0.05 | 0.09 | 0.11 | 0.08 | 0.07 | 0.69 | 1.61 | 2.84 | 1.71 | 1.47 | 6.70 |
| | TOKEN | **0.26** | **0.63** | **1.28** | **0.72** | **0.60** | **0.02** | **0.04** | **0.06** | **0.04** | **0.04** | **0.47** | **1.03** | **2.08** | **1.19** | **1.00** | **2.22** |

Table 2: **TOKEN significantly outperforms PARA-Drive in long-tail scenarios.**

**Results.** We present the quantitative evaluation of each model's scene understanding, object grounding, and planning abilities in Tab. 1 (only traj. L2 results are shown; full planning results are in App. E). Video-LLaMA and VILA-1.5 perform noticeably worse across all metrics, particularly in critical object grounding. This is because their encoders are optimized for visual-text alignment, and the limited driving QAs do not sufficiently develop their 3D understanding ability. Notably, Video-LLaMA and BEV-TOKEN share the same tokenization scheme and only differ in visual features, highlighting the value of visual features pre-trained with driving-related tasks. The benefits of pre-training features using embodied-related tasks are also observed in MM-LLM for robot manipulations [33]. Interestingly, TOKEN significantly outperforms BEV-TOKEN in all tasks, even though they tokenize the same visual features. This demonstrates the benefits of object-centric tokenization.

*Takeaway 1: Object-centric tokenization with pre-trained driving-aware features enables better grounding, reasoning, and planning performance in low-data regimes.*

## 5.2 Generalization in Long-tail Driving Scenarios

We evaluate the planning performance of TOKEN against PARA-Drive in long-tail events. TOKEN uses the PARA-Drive as the scene-tokenizer, thus they share the same visual encoder and scene information, and only differ in the planner structure.

**Long-tail Events Construction.** We manually inspected the NuScenes dataset and identified the following long-tail scenarios for evaluation, each representing less than 1% of the training data: 1) executing 3-point turns; 2) resuming motion after a full stop; 3) overtaking parked cars through the oncoming lane; and 4) navigating around construction sites. More details can be found in App. F.

**Quantitative Result.** In Tab. 2, we summarize the quantitative evaluation for each long-tail scenario. TOKEN significantly outperforms PARA-Drive in terms of the quality and safety of predicted plans. Specifically, TOKEN predicts more accurate turning maneuvers in 3-point turns (60% reduction in heading L2 Ave$_{all}$), more effective motions after yielding (28% reduction in longitudinal weighted L2 Ave$_{all}$), and safer plans when navigating around blocking vehicles and construction sites (100% and 67% collision rate reductions in the overtake and construction zone scenarios, respectively).

**Qualitative Examples.** In our qualitative analysis, TOKEN consistently outperforms PARA-Drive. During a 3-point turn (Fig. 4), TOKEN accurately generates the turning maneuver while PARA-Drive struggles. In Fig. 5, after yielding to pedestrians, TOKEN predicts a forward motion, whereas PARA-Drive remains stationary. In the opposite direction overtake scenario (Fig. 6), TOKEN predicts motions that avoid collisions and attempt to return to its lane post-overtake, unlike PARA-Drive, which predicts unsafe motions. Finally, TOKEN generates plans that effectively steer around the construction zone, while PARA-Drive predicts motions that result in collisions or deviations from the lane (Fig. 7). More detailed analysis can be found in App. G.

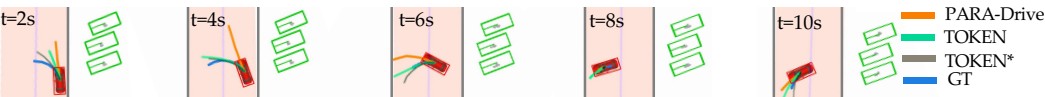

**Prompt:** The autonomous vehicle needs to make a left 3-point turn. What objects are important for the autonomous vehicle's planning? What are these objects, their (x,y) locations? and how should the autonomous vehicle interact with them? Please plan the autonomous vehicle's 3s future trajectory using 6 waypoints, one every 0.5 seconds.

**TOKEN:** There is no important object for the autonomous vehicle's planning. The autonomous vehicle should make a sharp left turn, and its 3s future trajectory is…[omitted]

Figure 4: Planning visualization in the 3-point U-turn scenario (zero-shot performance). TOKEN* denotes a model variant trained with additional synthetic data augmentation (few-shot performance).

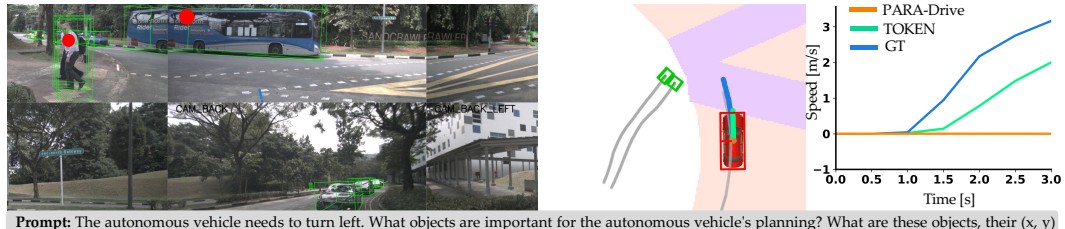

**Prompt:** The autonomous vehicle needs to turn left. What objects are important for the autonomous vehicle's planning? What are these objects, their (x, y) locations, and how should the autonomous vehicle interact with them? Please plan…[omitted]

**TOKEN:** There are 2 important objects: pedestrian at (-5.2, 5.4), bus at (-4.4, 16.3). It should continue to drive, accelerate and steer slightly to left…[omitted]

Figure 5: Planning performance visualization. The red dots in the left plot represent the identified critical objects. The middle plot visualizes the predicted motions. The right plot shows the speed plan inferred from the predicted motions. Note that TOKEN enables the ego vehicle to resume motion after a full stop.

Since 3-point turn is a zero-shot generalization scenario, we evaluate TOKEN's few-shot learning ability by generating synthetic 3-point turn motions and augmenting the training dataset with 40 samples (0.12% of the training data). The resulting model TOKEN*'s predicted motions (dark gray lines in Fig. 4) show qualitative improvement and better alignment with a 3-point turn. This example demonstrates the data efficiency of adapting LLM-based planners to new concepts.

*Takeaway 2: By leveraging the LLM's common-sense reasoning capabilities, TOKEN predicts more accurate plans in long-tail scenarios in which traditional end-to-end planning methods struggle.*

### 5.3 On the Value of Alignment

We compare TOKEN with the SOTA LLM-based planner Agent-Driver [5]. Agent-Driver queries text-based scene information using various tools and then fine-tunes GPT-3.5 into a motion planner. Our evaluation (Tab. 3) shows that TOKEN and Agent-Driver have similar overall performance, but TOKEN significantly outperforms agent-driver in long-tail scenarios with a much smaller model and less privileged information (details in App. H). We hypothesize that evoking common-sense reasoning in a LLM-powered planner requires proper alignment rather than just a larger LLM backbone. Motivated by this finding, we further conduct a detailed ablation study to shed more light on this.

| Split | Method | Traj L2 (m) ↓ | | | | | Heading L2 (rad) ↓ | | | | | Lon. weighted traj L2 (m) ↓ | | | | | Collision (%) ↓ |
|---|---|---|---|---|---|---|---|---|---|---|---|---|---|---|---|---|---|
| | | 1s | 2s | 3s | Ave$_{123s}$ | Ave$_{all}$ | 1s | 2s | 3s | Ave$_{123s}$ | Ave$_{all}$ | 1s | 2s | 3s | Ave$_{123s}$ | Ave$_{all}$ | Ave$_{all}$ |
| Val | Agent-driver | **0.23** | **0.69** | 1.52 | **0.81** | **0.67** | 0.03 | 0.06 | 0.11 | 0.06 | 0.06 | **0.43** | **1.27** | 2.78 | **1.49** | **1.23** | **0.13** |
| | TOKEN | 0.26 | 0.71 | **1.47** | 0.81 | 0.68 | **0.02** | **0.04** | **0.06** | **0.04** | **0.03** | 0.50 | 1.32 | **2.73** | 1.52 | 1.27 | 0.15 |
| Long-tail | Agent-Driver | 0.38 | 1.31 | 2.93 | 1.54 | 1.26 | 0.07 | 0.21 | 0.35 | 0.21 | 0.18 | **0.64** | 2.26 | 4.78 | 2.56 | 2.11 | 2.67 |
| | TOKEN | **0.26** | **0.81** | **1.77** | **0.95** | **0.78** | **0.05** | **0.10** | **0.18** | **0.11** | **0.09** | 0.50 | **1.47** | **3.09** | **1.69** | **1.40** | **0.35** |

Table 3: **Quantitative comparison with an LLM-based planner - Agent-Driver.**

| Split | Alignment representation | reasoning | Traj L2 (m) ↓ | | | | | Heading L2 (rad) ↓ | | | | | Lon. weighted traj L2 (m) ↓ | | | | | Collision (%) ↓ |
|---|---|---|---|---|---|---|---|---|---|---|---|---|---|---|---|---|---|---|
| | | | 1s | 2s | 3s | Ave$_{123s}$ | Ave$_{all}$ | 1s | 2s | 3s | Ave$_{123s}$ | Ave$_{all}$ | 1s | 2s | 3s | Ave$_{123s}$ | Ave$_{all}$ | Ave$_{all}$ |
| Val | x | x | 0.34 | 0.85 | 1.67 | 0.96 | 0.80 | 0.05 | 0.08 | 0.08 | 0.07 | 0.08 | 0.66 | 1.62 | 3.14 | 1.81 | 1.52 | 0.22 |
| | ✓ | x | 0.28 | 0.75 | 1.55 | 0.86 | 0.72 | 0.03 | 0.05 | 0.08 | 0.05 | 0.04 | 0.5 | 1.32 | 2.72 | 1.51 | **1.26** | 0.19 |
| | x | ✓ | **0.20** | 0.75 | 1.89 | 0.95 | 0.76 | 0.03 | 0.06 | 0.11 | 0.07 | 0.7 | **0.35** | 1.38 | 3.48 | 1.74 | 1.41 | 0.24 |
| | ✓ | ✓ | 0.26 | **0.71** | **1.47** | **0.81** | **0.68** | **0.02** | **0.04** | **0.06** | **0.04** | **0.03** | 0.50 | **1.32** | **2.73** | 1.52 | 1.27 | **0.15** |
| Long-tail | x | x | 0.35 | 1.10 | 2.39 | 1.29 | 1.06 | 0.12 | 0.19 | 0.24 | 0.18 | 0.18 | 0.60 | 1.81 | 3.87 | 2.10 | 1.74 | 0.61 |
| | ✓ | x | 0.32 | 0.99 | 2.09 | 1.13 | 0.94 | 0.11 | 0.17 | 0.21 | 0.16 | 0.15 | 0.60 | 1.78 | 3.69 | 2.02 | 1.68 | 0.52 |
| | x | ✓ | 0.36 | 1.09 | 2.26 | 1.24 | 1.03 | 0.14 | 0.19 | 0.23 | 0.18 | 0.17 | 0.64 | 1.93 | 3.98 | 2.18 | 1.82 | 0.47 |
| | ✓ | ✓ | **0.26** | **0.81** | **1.77** | **0.95** | **0.78** | **0.05** | **0.10** | **0.18** | **0.11** | **0.09** | 0.50 | **1.47** | **3.09** | **1.69** | **1.40** | **0.35** |

Table 4: **Value of alignment.** Both the pre-training representation alignment and structured reasoning alignment are necessary to leverage the LLM's common sense reasoning ability.

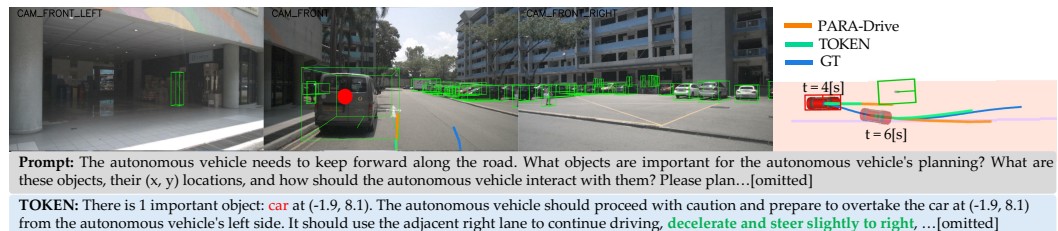

**Prompt:** The autonomous vehicle needs to keep forward along the road. What objects are important for the autonomous vehicle's planning? What are these objects, their (x, y) locations, and how should the autonomous vehicle interact with them? Please plan…[omitted]

**TOKEN:** There is 1 important object: car at (-1.9, 8.1). The autonomous vehicle should proceed with caution and prepare to overtake the car at (-1.9, 8.1) from the autonomous vehicle's left side. It should use the adjacent right lane to continue driving, **decelerate and steer slightly to right**, …[omitted]

Figure 6: Planning performance visualization. The right plot visualizes the predicted motion plans at two consecutive time steps. TOKEN enables the ego vehicle to decelerate when approaching the obstacle and drive back to the original lane after overtaking.

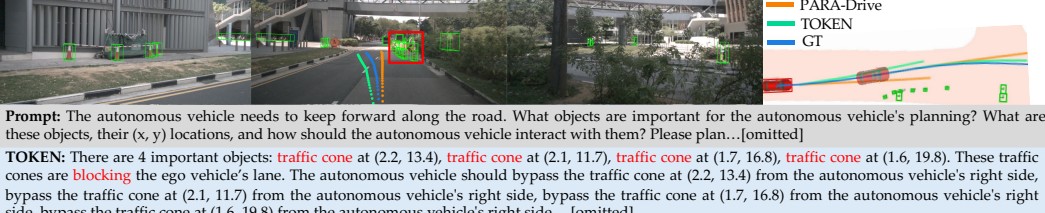

**Prompt:** The autonomous vehicle needs to keep forward along the road. What objects are important for the autonomous vehicle's planning? What are these objects, their (x, y) locations, and how should the autonomous vehicle interact with them? Please plan…[omitted]

**TOKEN:** There are 4 important objects: traffic cone at (2.2, 13.4), traffic cone at (2.1, 11.7), traffic cone at (1.7, 16.8), traffic cone at (1.6, 19.8). These traffic cones are blocking the ego vehicle's lane. The autonomous vehicle should bypass the traffic cone at (2.2, 13.4) from the autonomous vehicle's right side, bypass the traffic cone at (2.1, 11.7) from the autonomous vehicle's right side, bypass the traffic cone at (1.7, 16.8) from the autonomous vehicle's right side, bypass the traffic cone at (1.6, 19.8) from the autonomous vehicle's right side …[omitted]

Figure 7: Planning performance visualization. The red rectangle denotes the identified critical traffic cones. The right plot visualizes the predicted motions at two constitutive time steps. TOKEN instructs the ego vehicle to steer and bypass the traffic cones from the ego vehicle's right side.

**Ablation Study.** We test three variants of TOKEN to investigate the value of alignment: 1) no representation alignment nor reasoning alignment - treating TOKEN as a traditional end-to-end model and training it to directly predict the ego vehicle's motion plan; 2) with representation alignment but no reasoning alignment; 3) without representation alignment but with reasoning alignment.

**Results.** We summarize the quantitative evaluation results over the evaluation set and all long-tail scenarios in Tab. 4. Our evaluation shows that without the pre-training stage, the performance variation is not significant in both the validation and long-tail splits. This indicates the necessity of aligning the sensory token's embedding space with the text embedding space. Interestingly, with representation alignment pre-training, additional reasoning process alignment does not significantly benefit the entire validation set but significantly improves the accuracy and safety of predicted motion plans in the long-tail split. This suggests that proper reasoning process alignment is essential to unlock the common-sense reasoning ability encoded in the LLM backbone. Qualitative comparisons can be found in the App. I.

*Takeaway 3: Representation and structured reasoning process alignments are important for evoking the common-sense reasoning ability of the LLM backbone for planning.*

We include additional results in App. J to demonstrate the performance improvement brought by the additional HD-map modality, the impact of interaction mode (as opposed to simple behavior mode) in the reasoning alignment, and further highlight the few-shot learning ability of TOKEN.

## 6 Limitation & Future Work

A strength and limitation of TOKEN is using a pre-trained and frozen end-to-end driving model as the scene tokenizer. This allows for extracting informative tokens and controlled experiments. However, TOKEN's performance is tightly coupled with the quality of the pretrained tokenizer. We show a failure case in App. K in which the critical object is not detected by the tracking querying transformer in PARA-Drive. Consequently, TOKEN is unable to understand the scene and generates incorrect behavior. Further work will focus on co-training PARA-Drive to leverage the knowledge within the LLM to improve the scene tokenizer. Another limitation is outputting the trajectory plan as text, with each digit as a separate token. This increases computational expense and complicates motion generation, leading to misalignment between predicted behavior and the motion plan (e.g., TOKEN predicts the correct behavior but incorrect motion in Fig. 6). Future work will focus on quantizing motions into discrete LLM-understandable tokens [34] or using a dedicated trajectory decoder [35]. More discussions about inference speed and closed-loop evaluations are in App. L.

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

# Supplementary Materials

## A    Extended Related Work

**Object-Centric Representation**. Existing MM-LLMs often utilize Vision Transformers (ViTs) to tokenize visual inputs into latent tokens. These tokens, resembling a static grid over the inputs, are unstructured and pose challenges for LLMs to interpret in embodied reasoning tasks [33]. To address this, prior work [20] employs vectors of symbolic representations generated by a state estimation module to encode scene information, which are then used as tokens for the LLM. Our work is closely related to [33] and [36], which use object scene representation transformers [37] to extract object-level latent tokens. However, unlike [33] and [36], our paper focuses on the autonomous driving domain and use an end-to-end driving model to tokenize the scene.

## B    Dataset Construction

### B.1    Examples of the QAs

In Fig. 8, we illustrate examples of QAs used in training TOKEN.

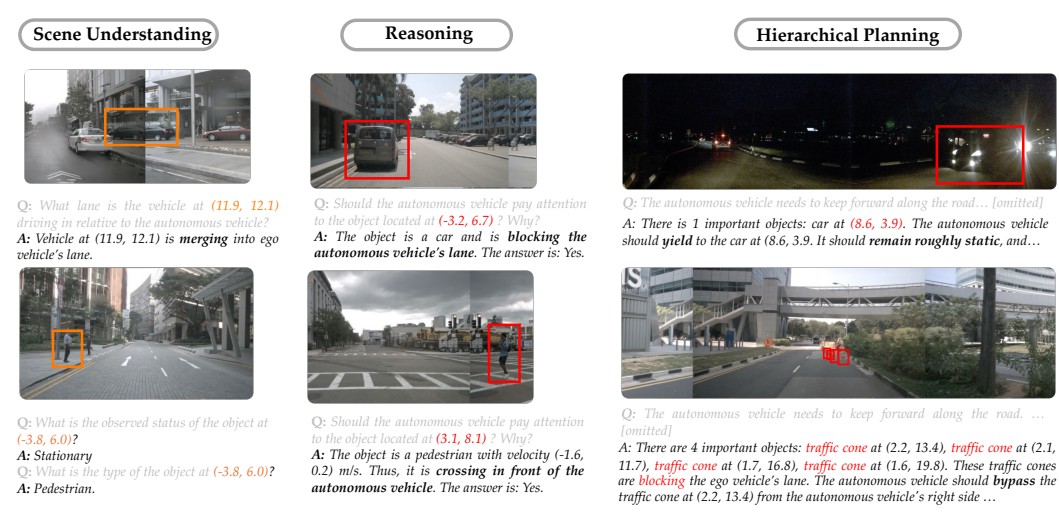

Figure 8: Examples of perception, reasoning, and planning QAs.

### B.2    Road-Level Navigation Signal

Previous works on VLM/LLM for autonomous vehicle planning often prompt the model with a high-level command based on the relative position of the ground-truth ego trajectory, including "keep forward" and "turn right/left." However, these high-level commands are not only unrealistic but also simplify the planning problem by removing the need for behavior planning. Therefore, we re-labeled the NuScenes dataset to use road-level navigation signals as high-level commands, including "keep forward along the current road," "prepare to turn right/left at the next intersection," "turn right/left at the intersection," "left/right U-turn," and "left/right 3-point turn."

### B.3    Interaction Mode Labeling

We use a combination of heuristics and manual labeling to annotate the interactions between the ego vehicle and the other traffic agents. We first use two types of categorical modes to describe the lane-relationship between a traffic agent and the ego vehicle *(agent-ego lane mode)* and the relative motion between a traffic participant and the ego vehicle *(homotopy)*[35]. Agent-ego lane mode at a time step $t$ encodes the topology relationship between the ego's current lane and the traffic agent's

lane, including: *LEFT*, *RIGHT*, *AHEAD*, *BEHIND*, and *NOTON*, where *NOTON* describes that the traffic agent is not on any derivable lanes in the scene (e.g., a parked vehicle in a parking lot). To compute the agent-ego lane mode for each traffic agent, we follow [35] to first identify the lane on which each agent is located and then leverage the lane topology map to annotate the agent-ego lane mode. We project the agent's center to the lane polyline and use its relative position in the local Frenet frame to determine its lane association. Homotopies describe the relative motion between a pair of agents shown in the video, including: [*S*, *CW*, *CCW*] (*static, clockwise, counterclockwise*). Combining agent-ego lane mode, homotopy, agent ground truth state information, and scene context information (e.g., ego is located near an intersection) together, we can leverage heuristics to annotate the interaction. For example, within a 3-second horizon, a static object's agent-ego lane mode changes from *AHEAD*, to *LEFT*, to *BEHIND*, while the ego vehicle performs *RIGHT-LANE-CHANGE*, *KEEP-LANE*, then *LEFT-LANE-CHANGE*, indicating the ego vehicle overtakes that object from the ego vehicle's left side. Finally, we use human labelers to verify and correct interaction labels in the following categories: 1) bypass blocking traffic cones to navigate around a construction zone; 2) yield to pedestrians; 3) yield to vehicles; 4) overtake traffic agents via straddling the lane dividers; 5) overtake traffic agents via lane-change.

## C   Training Details

During pre-training ("representation alignment"), we disable LoRA and only train the adapter to enhance embedding space alignment between the scene and text tokens. We use only perception QAs (150k QAs) to train the adapter for 5 epochs with a learning rate of 5e-4. In reasoning fine-tuning ("reasoning alignment"), we train the adapter and LoRA together using the reasoning and planning QAs for 10 epochs with a learning rate of 1e-4. Finally, we train the adapter and LoRA together using just the planning QAs for another 10 epochs to maximize the model's performance on planning, maintaining the learning rate at 1e-4.

## D   Evaluation Protocol

In this section, we provide a detailed description about our evaluation protocol. In Section 5.1 of the main text, we introduce the three variants of trajectory L2 error (the overall, turning, and progress errors) and the collision rate used to evaluate the predicted motion plans. As noted in [2], different evaluation protocols used to compute these metrics can lead to significant metric variations. We use the same evaluation protocol as described in [2] with one exception: we exclude samples where any future motion is missing near the end of a sequence (the frame masking strategy described in [2]). Including these partially invalid samples would significantly lower the L2 errors, as the L2 errors of these invalid frames are set to zero.

## E   Additional Result: On the Value of Object-Centric Tokenization

In Tab. 5, we present the full quantitative evaluation of each model's performance in the planning task. We observe that TOKEN significantly outperforms all baselines across all planning metrics.

| Method | Traj L2 (m) ↓ | | | | | Heading L2 (rad) ↓ | | | | | Lon. weighted traj L2 (m) ↓ | | | | | Collision (%) ↓ |
|---|---|---|---|---|---|---|---|---|---|---|---|---|---|---|---|---|
| | 1s | 2s | 3s | Ave$_{123s}$ | Ave$_{all}$ | 1s | 2s | 3s | Ave$_{123s}$ | Ave$_{all}$ | 1s | 2s | 3s | Ave$_{123s}$ | Ave$_{all}$ | Ave$_{all}$ |
| Video-LLaMA | 0.27 | 1.72 | 6.34 | 3.01 | 2.39 | 0.06 | 0.14 | 0.20 | 0.13 | 0.13 | 0.56 | 3.36 | 9.20 | 4.36 | 3.52 | 2.64 |
| VILA-1.5 | 0.28 | 1.56 | 4.41 | 2.09 | 1.66 | 0.05 | 0.11 | 0.19 | 0.12 | 0.10 | **0.29** | 1.92 | 6.47 | 2.89 | 2.24 | 1.98 |
| BEV-TOKEN | 0.39 | 1.01 | 2.02 | 1.14 | 0.96 | 0.03 | 0.05 | 0.06 | 0.05 | 0.05 | 0.75 | 1.79 | 3.55 | 2.03 | 1.71 | 0.39 |
| TOKEN | **0.26** | **0.70** | **1.46** | **0.81** | **0.68** | **0.02** | **0.04** | **0.06** | **0.04** | **0.03** | 0.50 | **1.32** | **2.72** | **1.51** | **1.26** | **0.15** |

Table 5: **Planning performance evaluation.** TOKEN significantly outperforms baseline VLMs due to its use of driving-task pre-trained features and object-centric tokenization.

## F   Long-tail Events Construction

We manually inspected the NuScenes dataset and identified the following long-tail scenarios for evaluation, each representing less than 1% of the training data: 1) executing 3-point turns; 2) resum-

ing motion after a full stop; 3) overtaking parked cars through the oncoming lane; and 4) navigating around construction sites. We note that our design choice of extracting these long-tail driving scenarios is statistically valid with respect to the accessible dataset. As most open-sourced datasets down-sample the meta dataset and only provide selected driving segments, they are inevitably altered and deviate from the actual daily driving distribution. Nevertheless, the above scenarios we selected are still representative and challenge the traditional end-to-end driving

**Executing 3-point turns**, which has one scene (scene-0778, frame 6-30) in the evaluation split and 0 scenes in the training distribution. We extracted 25 key frames from the scene in which the ego vehicle is performing the 3-point U-turn for evaluation to remove the noise from other nominal behaviors in the scene.

**Resuming motion after a full-stop**, which includes 14 scenes in the training split and 8 scenes in the evaluation split. We extract key frames from each scene where the ground truth (GT) motion plan captures the acceleration behavior after the full stop. This results in 70 key frames in the training split (0.28% of the total training samples) and 40 key frames in the evaluation split. The scenes and key frames in the training split are: scene-0208 (frame 25-29), scene-1023 (frame 21–25), scene-0067 (frame 24-28), scene-0159 (frame 4-8), scene-0185 (frame 26-30), scene-0262 (frame 8-12), scene-0862 (frame 18-22), scene-0025 (frame 6-10) scene-0072 (frame 24-28), scene-0157 (frame 12-16), scene-0234 (frame 4-8), scene-0423 (frame 6-10), scene-0192 (frame 14-18), and scene-0657 (frame 12-16). The scenes and key frames in the evaluation split are: scene-0921 (frame 21-25), scene-0925 (frame 19-23), scene-0968 (frame 7-11), scene-0552 (13-17), scene-0917 (frame 24-28), scene-0221 (frame 11-15), scene-1064 (frame 21-25), and scene-0331 (frame 8-12).

**Overtaking parked cars through the oncoming lane**, which includes 14 scenes in the training split and 5 scenes in the evaluation split. We extracted key frames from each scene where the ground truth motion plan captures the ego vehicle steering into the adjacent lane to overtake a blocking object and then returning to its original lane. This results in 248 key frames in the training split (0.9% of the total training samples) and 102 key frames in the evaluation split. The scenes and key frames in the training split are: scene-0001 (frame 12-39), scene-0011 (frame 1-39), scene-0023 (frame 1-8), scene-0034 (frame 23-39), scene-0318 (frame 10-30), scene-0379 (frame 14-26), scene-0408 (frame 12-30), scene-0417 (frame 4-20), scene-0422 (frame 18-39), scene-0865 (frame 24-39), scene-1105 (frame 18-30), scene-1065 (frame 24-35), scene-0200 (frame 20-39), and scene-0752 (frame 10-28). The scenes and key frames in the evaluation split are: scene-0038 (frame 4-33), scene-0271 (frame 3-11), scene-0969 (frame 14-33), scene-0329 (frame 3-33), and scene-1065 (frame 24-35)

**Navigating around construction sites**. This common and challenging scenario requires the autonomous vehicle to actively change lanes to bypass a construction zone. Although there are two scenes (scene-0980 and scene-0535) in the training split, none exist in the evaluation split. Therefore, we moved one training scene (scene-0980, frames 16-30) to the evaluation split. We selected 15 key frames that capture the ego vehicle decelerating and steering into the adjacent lane to bypass the traffic cones.

## G   Detailed Qualitative Result

In this section, we provide an in-depth analysis of the qualitative results shown in Sec. 5.2.

**Executing a 3-point turn**. During a 3-point turn, a vehicle makes a sharp left turn, backs up, and then makes another left turn to complete the maneuver. In Fig. 4, we compare the motion plans from TOKEN and PARA-Drive. Despite receiving a "3-point turn" command, PARA-Drive predicts straight movements at $t = 2$s and $t = 4$s, likely due to the absence of such examples in its training set. In contrast, TOKEN understands the command and generates the correct turning behavior. When approaching the curb to stop and back up, both PARA-Drive and TOKEN predict forward motions at $t = 8$s, likely due to the lack of 3-point turn examples in the dataset. At $t = 10$s, when the vehicle has enough clearance, both models predict left-turn motions, with TOKEN's prediction more closely aligning with the ground truth.

**Overtaking parked cars through the oncoming lane.** Fig. 6 shows an example of qualitative comparison between the motion plan from TOKEN and PARA-Drive at two constitutive time steps. At $t = 5$s, PARA-Drive predicts a motion that collides with the blocking vehicle, while TOKEN instructs the ego vehicle to decelerate to avoid the object. Interestingly, although TOKEN correctly predicts in language that the ego vehicle should decelerate and steer to the right, the motion plan only reflects the deceleration behavior. We hypothesize that this is caused by insufficient data that helps the LLM associate low-level motion with mid-level behavior. When the ego vehicle straddles the lane divider and prepares to overtake, TOKEN instructs the ego vehicle to drive back to its original lane after overtaking, while PARA-Drive predicts a forward motion that straddles the lane divider.

**Navigating around construction sites.** Fig. 7 shows an example of qualitative comparison between the motion plan from TOKEN and PARA-Drive at two constitutive time steps. At $t = 8$s, TOKEN instructs the ego vehicle to steer and bypass the traffic cones from the ego vehicle's right side, while PARA-Drive predicts a motion that collides with the blocking traffic cones At $t = 12$s, TOKEN instructs the ego vehicle to steer to the right to keep forward along the current lane, while PARA-Drive predicts a motion that deviates from the lane center.

## H   Comparison with the SOTA LLM-based Planner - Agent-Driver

We compare TOKEN with the SOTA LLM-based planner Agent-Driver [5]. They have the following differences:

**Scene representation.** Agent-Driver queries text-based scene information using various tools (e.g., object detection, mapping, etc.) and uses the queried text-based information as an input prompt to instruct the LLM to plan the ego vehicle's motion. TOKEN tokenizes the scene into a few object-level tokens. This makes TOKEN more efficient in terms of information density per-token (e.g., TOKEN uses one token to encode an object's semantic, geometry, and dynamic information while the same information is tokenized into around 60 text tokens in Agent-Driver).

**LLM-backbone.** Agent-Driver fine-tunes the entire GPT3.5 while TOKEN uses LORA (with a rank of 64) to fine-tune LLaMA2.

**Chain-of-Thought Reasoning.** Both Agent-Driver and TOKEN utilize chain-of-thought reasoning to align the model's planning process. However, Agent-Driver uses a coarse reasoning process that instructs the LLM to directly generate the ego vehicle's discretized steering and acceleration command description based on the scene description, as opposed to TOKEN's more structured reasoning process that instructs the model to reason about the semantically meaningful effect of the objects (e.g., blocking the ego vehicle's path) and interaction plans (e.g., overtake).

We use the Agent-Driver's predictions from the official repository and show the overall quantitative evaluation in Tab. 3 of the main text. In Tab. 6, we show the detailed quantitative evaluation of each long-tail scenario. In addition, since Agent-Driver includes the ego-state information in the prompt, we train a variant of TOKEN (denoted as TOKEN$^+$) that also uses the ego-state information as input for planning (current velocity and current acceleration) and show the quantitative evaluation of TOKEN$^+$ in Tab. 6. We see that TOKEN and Agent-Driver have similar overall performance, but TOKEN significantly outperforms Agent-Driver in long-tail scenarios. Furthermore, when using ego-state information as input, TOKEN$^+$ significantly outperforms Agent-Driver in all splits.

## I   On the Value of Alignment - Qualitative Results

In Fig. 9, we show a few qualitative comparisons between TOKEN and a variant of TOKEN trained with representation alignment but without reasoning alignment. We can see that the predicted motion plans are more aligned the GT motion with reasoning process alignment.

| Split | Method | Traj L2 (m) ↓ | | | | | Heading L2 (rad) ↓ | | | | | Lon. weighted traj L2 (m) ↓ | | | | | Collision (%) ↓ |
|---|---|---|---|---|---|---|---|---|---|---|---|---|---|---|---|---|---|
| | | 1s | 2s | 3s | Ave$_{123s}$ | Ave$_{all}$ | 1s | 2s | 3s | Ave$_{123s}$ | Ave$_{all}$ | 1s | 2s | 3s | Ave$_{123s}$ | Ave$_{all}$ | Ave$_{all}$ |
| val | Agent-Driver | 0.23 | 0.68 | 1.50 | 0.80 | 0.66 | 0.79 | 0.85 | 0.90 | 0.85 | 0.80 | 0.43 | 1.26 | 2.75 | 1.48 | 1.22 | **0.13** |
| | TOKEN | 0.26 | 0.70 | 1.46 | 0.81 | 0.68 | 0.13 | 0.18 | 0.21 | 0.17 | 0.18 | 0.50 | 1.32 | 2.72 | 1.51 | 1.26 | 0.15 |
| | TOKEN $^+$ | **0.17** | **0.52** | **1.21** | **0.64** | **0.52** | **0.10** | **0.16** | **0.19** | **0.15** | **0.17** | **0.33** | **1.00** | **2.30** | **1.21** | **1.00** | **0.13** |
| 3-point turn | Agent-Driver | 0.38 | 1.31 | 2.93 | 1.54 | 1.26 | 0.20 | 0.74 | 1.23 | 0.72 | 0.63 | 0.64 | 2.26 | 4.78 | 2.56 | 2.11 | 8.67 |
| | TOKEN | 0.39 | 1.29 | 2.60 | 1.43 | 1.18 | 0.21 | 0.35 | 0.71 | 0.42 | 0.36 | 0.68 | 2.15 | 4.33 | 2.39 | 1.98 | 4.00 |
| | TOKEN $^+$ | **0.20** | **0.73** | **1.77** | **0.90** | **0.73** | **0.17** | **0.22** | **0.38** | **0.26** | **0.22** | **0.37** | **1.28** | **2.93** | **1.53** | **1.24** | **1.46** |
| Resume motion from full stop | Agent-Driver | 0.14 | 0.84 | 2.51 | 1.16 | 0.91 | 0.17 | 0.47 | 0.42 | 0.35 | 0.32 | 0.25 | 1.49 | 4.48 | 2.07 | 1.62 | 0.00 |
| | TOKEN | 0.13 | 0.70 | 1.58 | 0.80 | 0.65 | 0.09 | 0.24 | 0.31 | 0.22 | 0.19 | 0.24 | 1.24 | 2.66 | 1.38 | 1.13 | 0.00 |
| | TOKEN $^+$ | **0.06** | **0.43** | **1.27** | **0.59** | **0.46** | **0.05** | **0.13** | **0.17** | **0.12** | **0.10** | **0.11** | **0.78** | **2.30** | **1.06** | **0.84** | 0.00 |
| Overtake | Agent-Driver | 0.27 | 0.89 | 2.07 | 1.08 | 0.88 | 0.05 | 0.13 | 0.24 | 0.14 | 0.12 | 0.47 | 1.46 | 3.37 | 1.77 | 1.45 | 0.77 |
| | TOKEN | 0.29 | 0.77 | 1.63 | 0.90 | 0.74 | 0.04 | 0.07 | **0.11** | 0.07 | 0.09 | 0.53 | 1.36 | 2.86 | 1.58 | 1.31 | 0.19 |
| | TOKEN $^+$ | **0.15** | **0.46** | **1.04** | **0.55** | **0.46** | **0.02** | **0.07** | 0.13 | **0.07** | **0.06** | **0.29** | **0.83** | **1.75** | **0.95** | **0.80** | **0.00** |

Table 6: **Quantitative comparison with an LLM-based planner - Agent-Driver.** TOKEN significantly outperforms Agent-Driver in long-tail scenarios. TOKEN$^+$ denotes a variant of TOKEN that uses ego-state as input, similar to Agent-Driver.

# J  Additional Results

## J.1  TOKEN with HD-map Information

In the main text, we use multi-view video as the sensory input. In this section, we include HD-map as an additional input to evaluate the performance of TOKEN. We utilize the CTT encoder described in [35] to fuse each traffic agent's past state history with each lane's ground truth center line and produce a traffic agent token as an additional token for each traffic object. We use TOKEN $^{+\text{map}}$ to denote the variant of TOKEN with HD-map information and show its quantitative evaluation in Tab. 7. We can see that the additional map information and the past state history significantly improve the planning performance in both evaluation split and long-tail scenarios.

| Split | Method | Traj L2 (m) ↓ | | | | | Heading L2 (rad) ↓ | | | | | Lon. weighted traj L2 (m) ↓ | | | | | Collision (%) ↓ |
|---|---|---|---|---|---|---|---|---|---|---|---|---|---|---|---|---|---|
| | | 1s | 2s | 3s | Ave$_{123s}$ | Ave$_{all}$ | 1s | 2s | 3s | Ave$_{123s}$ | Ave$_{all}$ | 1s | 2s | 3s | Ave$_{123s}$ | Ave$_{all}$ | Ave$_{all}$ |
| Val | TOKEN | 0.26 | 0.71 | 1.47 | 0.81 | 0.68 | **0.02** | 0.04 | 0.06 | 0.04 | 0.03 | 0.50 | 1.32 | 2.73 | 1.52 | 1.27 | 0.15 |
| | TOKEN $^{+\text{map}}$ | **0.15** | **0.42** | **1.18** | **0.58** | **0.51** | **0.02** | **0.02** | **0.06** | **0.03** | **0.03** | **0.33** | **0.92** | **2.11** | **1.12** | **0.97** | **0.08** |
| Long-tail | TOKEN | 0.26 | 0.81 | 1.77 | 0.95 | 0.78 | 0.05 | 0.10 | 0.18 | 0.11 | 0.09 | 0.50 | 1.47 | 3.09 | 1.69 | 1.40 | 0.35 |
| | TOKEN $^{+\text{map}}$ | **0.20** | **0.71** | **1.37** | **0.76** | **0.64** | **0.03** | **0.08** | **0.11** | **0.07** | **0.05** | **0.37** | **1.18** | **2.73** | **1.43** | **1.27** | **0.31** |

Table 7: Quantitative performance of TOKEN $^{+\text{map}}$, a variant of TOKEN with HD-map information.

## J.2  Ablation on the Effect of Structured Reasoning Process Alignment.

One of the unique features of TOKEN is that it reasons about semantically meaningful interactions with identified critical objects (e.g., bypassing the blocking traffic cones). We hypothesize that this structured reasoning process supervision enhances the model's planning performance by encouraging the model to understand the interactions between the ego vehicle and other traffic objects, aligning more closely with how an expert reasons in the real world. To ablate the effect of structured reasoning process alignment, we removed it from the planning QAs' answer labels and used a similar chain-of-thought reasoning and task planning method as in Agent-Driver (i.e., instructing the LLM to generate the steering and acceleration command description first, followed by the motion plan) to TOKEN (denoted as TOKEN $^{-\text{interact.}}$). We show the evaluation result in Tab. 8. We can see that the structured reasoning process alignment improves the planning performance in both the evaluation set and the long-tail scenarios.

| Split | Method | Traj L2 (m) ↓ | | | | | Heading L2 (rad) ↓ | | | | | Lon. weighted traj L2 (m) ↓ | | | | | Collision (%) ↓ |
|---|---|---|---|---|---|---|---|---|---|---|---|---|---|---|---|---|---|
| | | 1s | 2s | 3s | Ave$_{123s}$ | Ave$_{all}$ | 1s | 2s | 3s | Ave$_{123s}$ | Ave$_{all}$ | 1s | 2s | 3s | Ave$_{123s}$ | Ave$_{all}$ | Ave$_{all}$ |
| Val | TOKEN $^{-\text{interact.}}$ | 0.28 | 0.77 | 1.54 | 0.86 | 0.75 | **0.02** | **0.03** | 0.08 | **0.04** | 0.04 | 0.52 | 1.39 | 2.82 | 1.58 | 1.31 | 0.17 |
| | TOKEN | **0.26** | **0.71** | **1.47** | **0.81** | **0.68** | **0.02** | 0.04 | **0.06** | **0.04** | **0.03** | **0.50** | **1.32** | **2.73** | **1.52** | **1.27** | **0.15** |
| Long-tail | TOKEN $^{-\text{interact.}}$ | 0.33 | 1.01 | 2.06 | 1.13 | 0.92 | 0.09 | 0.15 | 0.19 | 0.14 | 0.12 | 0.58 | 1.82 | 3.66 | 2.02 | 1.65 | 0.59 |
| | TOKEN | **0.26** | **0.81** | **1.77** | **0.95** | **0.78** | **0.05** | **0.10** | **0.18** | **0.11** | **0.09** | **0.50** | **1.47** | **3.09** | **1.69** | **1.40** | **0.35** |

Table 8: Quantitative performance of TOKEN $^{-\text{interact.}}$, a variant of TOKEN that uses a similar chain-of-thought reasoning as Agent-Driver does.

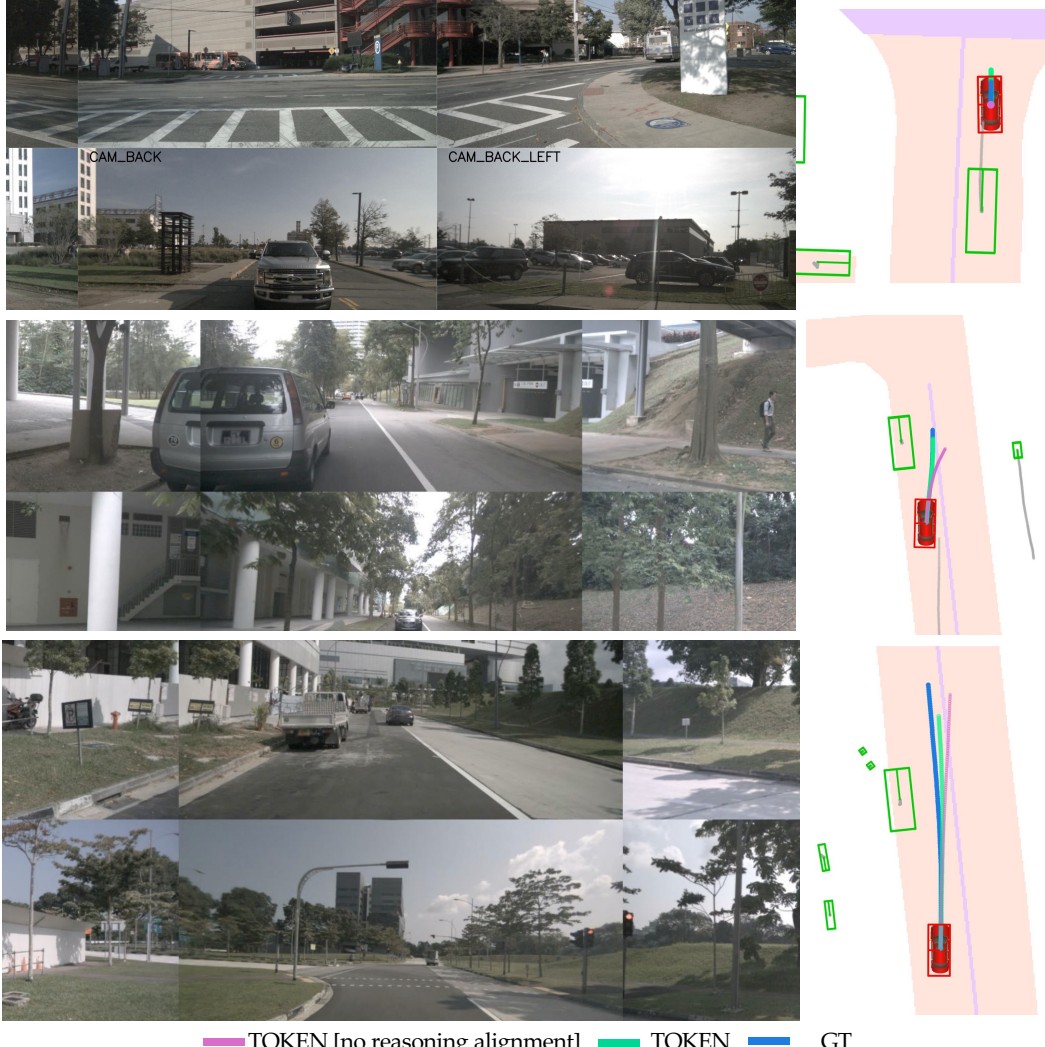

TOKEN [no reasoning alignment] ▬ TOKEN ▬ GT

Figure 9: Qualitative comparison between TOKEN and a variant of TOKEN trained with representation alignment but without reasoning alignment. With reasoning process alignment, TOKEN is able to instruct the ego vehicle to resume motion after a full-stop (top figure) and safely overtake static obstacles (middle and bottom figures).

## J.3 Few-Shot Learning.

To stress test TOKEN's few shot learning ability, we further remove 50% long-tail scenes from the training split and re-train PARA-drive and TOKEN and compare their performance. In Tab. 9, we show the quantitative evaluation result in long-tail scenarios. We see that TOKEN only degrades slightly as opposed to PARA-Drive's significant performance degradation. For example, Traj L2 $Ave_{all}$ of PARA-Drive is degraded by 24% while TOKEN only experiences 9% degradation with 50% long-tail scenes removed. The results indicate the superior few-shot learning ability of TOKEN.

| Split | Method | Traj L2 (m) ↓ | | | | | Heading L2 (rad) ↓ | | | | | Lon. weighted traj L2 (m) ↓ | | | | | Collision (%) ↓ |
|---|---|---|---|---|---|---|---|---|---|---|---|---|---|---|---|---|---|
| | | 1s | 2s | 3s | Ave123s | Aveall | 1s | 2s | 3s | Ave123s | Aveall | 1s | 2s | 3s | Ave123s | Aveall | Aveall |
| Long-tail (full long-tail training set) | PARA-Drive | **0.26** | 0.96 | 2.16 | 1.13 | 0.91 | 0.09 | 0.20 | 0.34 | 0.21 | 0.18 | 0.47 | 1.61 | 3.51 | 1.86 | 1.57 | 0.51 |
| | TOKEN | **0.26** | **0.81** | **1.77** | **0.95** | **0.78** | **0.05** | **0.10** | **0.18** | **0.11** | **0.09** | **0.50** | **1.47** | **3.09** | **1.69** | **1.40** | **0.35** |
| Long-tail (with 50% long-tail training scenes removed) | PARA-Drive | 0.51 | 1.17 | 2.30 | 1.32 | 1.12 | 0.09 | 0.17 | 0.29 | 0.18 | 0.16 | 0.76 | 1.96 | 3.93 | 2.22 | 1.86 | 0.89 |
| | TOKEN | **0.28** | **0.86** | **1.83** | **0.99** | **0.85** | **0.06** | **0.15** | **0.24** | **0.15** | **0.13** | **0.56** | **1.66** | **3.29** | **1.84** | **1.62** | **0.41** |

Table 9: Planning performance of TOKEN and PARA-Drive with 50% long-tail training scenes removed.

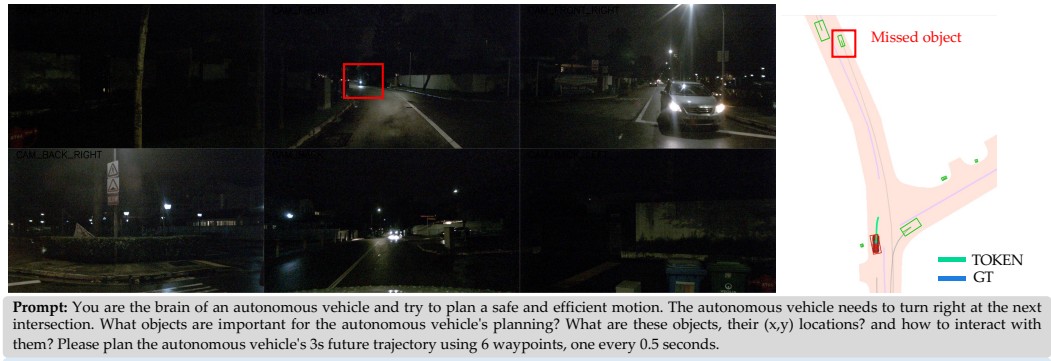

**Prompt:** You are the brain of an autonomous vehicle and try to plan a safe and efficient motion. The autonomous vehicle needs to turn right at the next intersection. What objects are important for the autonomous vehicle's planning? What are these objects, their (x,y) locations? and how to interact with them? Please plan the autonomous vehicle's 3s future trajectory using 6 waypoints, one every 0.5 seconds.

**TOKEN:** There is no important object for the autonomous vehicle's planning. The autonomous vehicle should keep forward, and its 3-second future trajectory is…[omitted]

Figure 10: Failure mode example: The critical object (an oncoming motorcycle annotated by the red rectangle) is not detected by TOKEN's scene tokenizer, resulting in a dangerous motion plan.

## K    Failure Mode

One limitation of TOKEN is using a pre-trained and frozen PARA-Drive model as the scene tokenizer, which makes the TOKEN's performance tightly coupled with the quality of the pretrained tokenizer. In Fig. 10, we illustrate a failure mode where the critical object (the motorcycle) is not detected by the tracking querying transformer in PARA-Drive. Consequently, TOKEN assumes the road is clear to proceed and fails to generate a motion that yields to the oncoming motorcycle. Further work will focus on co-training PARA-Drive to leverage the knowledge within the LLM to improve the scene tokenizer.

## L    Extended Limitation & Future Work

TOKEN currently is not validated in closed-loop evaluations. We think that open-loop metrics are still valuable, and enable us to efficiently validate the model in a controlled environment free from the noises brought by other components in closed-loop evaluations (e.g., agents reactivity, motion optimization, and control). To mitigate the bias induced by general open-loop metrics, we specifically evaluate our planning metrics using key frames that reflect the critical behavior/decision changes. We will investigate TOKEN's reasoning and planning performance with closed-loop simulation in our future work. Last but not least, one of the limitations of deploying LLM-based systems onboard is their slow inference speed. In our current setup, TOKEN takes approximately 1.3 seconds to generate mid-level behavior and around 1.8 seconds to generate the motion plan without any speed optimization using an A100 GPU. One practical deployment solution is to use TOKEN solely as a runtime decision-level monitor: instead of predicting the full motion plan, it could predict route-conditioned mid-level behavior (e.g., bypassing a construction zone) at a much slower frequency to inform the traditional motion planner. Although there are many ongoing efforts aimed at improving LLM inference speed through quantization and caching, the large model size still makes it infeasible for onboard deployment. One direction we are particularly interested in is distillation. In our future work, we are very excited about distilling the 7B TOKEN model into a much smaller, lightweight model that can retain the benefits of common-sense reasoning in long-tail events while being more suitable for onboard deployment.

