# OpenReview forum: "Tokenize the World into Object-level Knowledge to Address Long-tail Events in Autonomous Driving"
_robot-learning.org/CoRL/2024/Conference — CoRL 2024_

### Official Review · Reviewer_gVTX · 2024-07-11
**TOKENization for addressing long-tail scenarios in autonomous vehicles**

**Originality:** 4
**Technical Quality:** 4
**Clarity Of Presentation:** 3
**Potential Impact:** 3
**Recommendation:** 3
**Confidence:** 4

**Review:**

Strengths: The paper proposes an adapter to align the latent object-level token embedding space generated by, e.g., PARA-Drive, and the text embedding space generated by, e.g., LLaMA-2-7B for the subsequent behavior planning task.

The paper addresses long-tail scenarios with a data scarcity problem and achieves promising performance in few-shot generalization scenarios, outperforming existing frameworks in terms of trajectory L2 error and collision rates in long-tail scenarios.

Weaknesses:

The performance of the model TOKEN highly depends on the pre-trained tokenizer.

The model has limited performance measured in longitudinal-weighted L2 distance for a 3-point turn, as shown in Table 2.

Major concerns:

PARA-Drive’s planning performance (shaded in grey in Table 1) is evidently better than TOKEN, but the authors did not elaborate on that.

In Table 6, the visualization shows some fragmentation in the predictions of TOKEN and PARA-Drive. The authors should comment on that.

Will the constructed QA dataset be released to further facilitate research for the long-tail driving scenarios?

**Quality Of The Limitations Section:**

3

**Questions For Rebuttal:**

Elaboration on the longitudinal-weighted L2 distance for a 3-point turn, as shown in Table 2, and the PARA-Drive’s planning performance (shaded in grey in Table 1).

Why there is some fragmentation in the predictions of TOKEN and PARA-Drive?

**Robotics Focus:**

3

**Summary Of Paper:**

This paper proposes object-centric tokenization (TOKEN) to extract informative and structured information to address long-tail scenarios for the planning ability of autonomous vehicles. These long-tail scenarios include executing 3-point turns, resuming motion after a full stop, overtaking parked cars through the oncoming lane, and navigating construction sites.

**Summary Of Recommendation:**

The paper shows a good potential to address long-tail scenarios for autonomous driving, but some revisions are needed to fix the concerns.

---

### Official Review · Reviewer_vv3a · 2024-07-20
**TOKEN utilizes the pre-trained end-to-end model and MM-LLM for motion planning, and shows strong generalization ability in long-tail scenarios. I raise concerns about the novelty and only open-loop evaluation (on static agent datasets) instead of closed-loop evaluation.**

**Originality:** 2
**Technical Quality:** 3
**Clarity Of Presentation:** 4
**Potential Impact:** 2
**Recommendation:** 2
**Confidence:** 5

**Review:**

Strength:
1. TOKEN stress the important of object-centric tokenization alignment for MM-LLM planning task.
2. Without the open-set sensing ability of MM-LLM mode, TOKEN shows strong generalization ability through representation and reasoning alignment.

Weakness:
1. The similar idea is given in LMDriver[1], which also use an end-to-end backbone as the scene tokenizer. TOKEN is more of a combination of  LMDriver and Driving with LLMs[2].
2. TOKEN demonstrates its superior grounding, reasoning, and planning capabilities in a low-data regime. However, a lot of end-to-end data is required to pre-train the end-to-end backbone PARA-Driver.
3. The trajectory L2 performance is not better than end-to-end model PARA-Driver, although it has strong generalization ability. It means that TOKEN can not handle common scenarios well enough like PARA-driver.
4. Motion planning needs to be checked in a closed-loop system because, as shown in [3,4], the outcomes of open-loop and closed-loop situations don't always match up. There are also some public datasets that can be used for closed-loop tests, like Nuplan, Bench2Drive, MetaDrive, and more. I believe that getting closed-loop results would greatly improve the credibility and usefulness of the work.
5. Do we really need billions of QAs for MM-LLM planning？ I think CarLLaVA[5] provides an interesting try.
6. The inference time and potential possibility to deploy in real vehicles for TOKEN should be analyzed.

[1] LMDrive: Closed-Loop End-to-End Driving with Large Language Models，CVPR， 2024.
[2] Driving with LLMs: Fusing Object-Level Vector Modality for Explainable Autonomous Driving， ICRA，2024
[3] Parting with Misconceptions about Learning-based Vehicle Motion Planning. CoRL, 2023
[4] Is Ego Status All You Need for Open-Loop End-to-End Autonomous Driving? CVPR, 2024.
[5] CarLLaVA: Vision language models for camera-only closed-loop driving, 2024.

**Quality Of The Limitations Section:**

2

**Questions For Rebuttal:**

I hope to see closed-loop results; please refer to the "Weaknesses" section for more details.

**Robotics Focus:**

2

**Summary Of Paper:**

In this paper, the pre-trained end-to-end backbone is used as a scene tokenizer for the MM-LLM planning task. The representation and structured reasoning process alignments are important for evoking the common-sense reasoning ability of the LLM backbone for planning. The open-loop performances demonstrate the superiority of Token in long-tail scenarios.

**Summary Of Recommendation:**

The overall text flows smoothly and is easy to understand. However, the novelty and the closed-loop performances should be further considered.

---

### Official Review · Reviewer_Ybio · 2024-07-20
**Interesting question but not rigorous enough**

**Originality:** 3
**Technical Quality:** 3
**Clarity Of Presentation:** 2
**Potential Impact:** 3
**Recommendation:** 3
**Confidence:** 4

**Review:**

Pros:
- Tons of experiments are conducted to support multiple claims
- Tokenizing scene is an interesting and fundamental question
- The observation about object-level tokenizing requires less data is useful

Questions:
- Can authors further justify how the long-tail dataset was built? If the reviewer understands correctly, the paper constructs its long-tail dataset by selecting scenarios from the NuScenes dataset that occur in less than 1% of the training data. However, this approach is flawed, as it primarily focuses on dataset frequency rather than real-world rarity or difficulty. For instance, resuming motion after a full stop is hard to consider as a rare and difficult scenario.
- Can you elaborate on your process for generating evaluation prompts? Are they created manually, automatically, or through some combination? How do you ensure that the prompts used for evaluation are as neutral as possible and do not inadvertently provide hints or bias toward certain types of responses?
- What is the metrics definition of scene understanding and critical object grounding? The meaning of the first two columns in Tab. 1 is unclear.
- Despite introducing LoRA making the approach beat SOTA, it also makes it hard to justify how much the proposed tokenization process contributes to the final result. Is it possible other LLM driving approaches without the tokenization process can also achieve similar performance by fine-tuning with LoRA and the authors' dataset?
- How well would the object-centric tokenization approach scale to more complex environments with a higher density of objects? Are there limitations to the number of objects that can be effectively tokenized and processed by the LLM?
- LLM-based planners may face challenges with planning latency, which is critical for real-time autonomous driving. What are your thoughts on addressing potential latency issues in such systems?
- LLM planners require carefully designed primitives to translate high-level decisions into executable vehicle commands. Designing effective primitives that bridge the gap between language outputs and low-level control can be challenging, especially for complex maneuvers. What are your thoughts on addressing the primitive design challenge for LLM-based autonomous driving systems?

**Quality Of The Limitations Section:**

2

**Questions For Rebuttal:**

See review

**Robotics Focus:**

3

**Summary Of Paper:**

One significant challenge of driving with LLMs is tokenizing scenes. Describing scenes with structured or natural language may cause loss or inaccuracy of information. The authors argue that embedding objects/scenes with an object-centric transformer and then aligning the modalities can better tokenize the scenes, thus providing better performance in downstream planning tasks in long-tail scenarios.

**Summary Of Recommendation:**

While the paper presents an innovative approach to scene tokenization for LLM-based autonomous driving, several key limitations undermine its overall impact. Further work is needed to address concerns about dataset construction, evaluation methodology, scalability, and real-world applicability before this method can be considered truly promising for autonomous driving systems.

---

### Author Rebuttal · Authors · 2024-08-10

We thank all the reviewers for their helpful comments and suggestions! We have updated our manuscript to incorporate the comments and suggestions.

8.14 Update: We added our statement about the closed-loop evaluation in the limitations and future work section of our updated paper.

---

### Decision · Program_Chairs · 2024-09-04

**Decision:**

Accept

**Comment:**

The paper "Tokenize the World into Object-level Knowledge to Address Long-tail Events in Autonomous Driving" introduces TOKEN, a Multi-Modal Large Language Model (MM-LLM) designed to improve autonomous vehicle planning in long-tail scenarios, or unusual and infrequent events, by tokenizing the world into object-level knowledge.

Strengths:
- Insight that object level tokenization needs less data is useful
- Aligns text level representations like Llama with object level representations
- Long tail scenario is a data hungry regime, achieves few shot generalization

Weaknesses:
- Performance highly depends on pretrained tokenizer, and the latter needs lots of data. Inference latency might be slower given larger models?
- Trajectory end to end performance not better than PARA-driver. This is a major weakness and is not well explained.
- No experiments on hardware

Reviewer Ybio initially gave a "Weak Accept" rating, questioning the construction of the long-tail dataset, the evaluation prompt generation process, and the scalability of the object-centric tokenization approach. The authors responded by providing additional details about their dataset construction methodology, explaining that it was statistically valid given the available data. They also clarified that evaluation prompts were manually generated to ensure neutrality. Regarding scalability, they acknowledged the current limit of 64 object tokens but suggested potential improvements like pre-filtering object tokens.

Reviewer vv3a initially recommended a "Weak Reject," raising concerns about the novelty of the approach, the lack of closed-loop evaluation, and the need for analysis of inference time and deployment possibilities. The authors addressed these concerns by highlighting the differences between their work and LMDriver, clarifying the training data used for the tokenizer, and discussing potential solutions for inference latency and onboard deployment. They also acknowledged the limitations of open-loop evaluation but emphasized its value for efficient model validation. The reviewer did not update their "Weak Reject" recommendation.

Reviewer gVTX provided a "Weak Accept" rating but pointed out the model's dependence on the pre-trained tokenizer, its limited performance in certain scenarios, and the lack of elaboration on PARA-driver's superior performance in some cases. The authors responded by discussing the possibility of co-training the tokenizer in future work, explaining the nuanced evaluation of motion plans using different L2 error variants, and addressing the comparison with PARA-driver in their general statement. They also committed to releasing their dataset and code.

Overall I recommend weak accept.